# Antimicrobial Properties of *Bacillus* Probiotics as Animal Growth Promoters

**DOI:** 10.3390/antibiotics12020407

**Published:** 2023-02-17

**Authors:** Charlie Tran, Darwin Horyanto, Dragana Stanley, Ian E. Cock, Xiaojing Chen, Yunjiang Feng

**Affiliations:** 1Griffith Institute for Drug Discovery (GRIDD), Griffith University, Brisbane, QLD 4111, Australia; charlie.tran@griffithuni.edu.au; 2Institute for Future Farming Systems, Central Queensland University, Rockhampton, QLD 4702, Australia; darwin.horyanto@bioproton.com (D.H.); d.stanley@cqu.edu.au (D.S.); 3Bioproton Pty Ltd., Brisbane, QLD 4110, Australia; wendy@bioproton.com; 4School of Environment and Science, Griffith University, Brisbane, QLD 4111, Australia; i.cock@griffith.edu.au

**Keywords:** animal feed, antimicrobials, *Bacillus*, omics, probiotics, spore-forming

## Abstract

Antibiotic growth promoters (AGPs) suppress the growth of infectious pathogens. These pathogens negatively impact agricultural production worldwide and often cause health problems if left untreated. Here, we evaluate six *Bacillus* strains (BPR-11, BPR-12, BPR-13, BPR-14, BPR-16 and BPR-17), which are known for their ability to survive harsh environmental conditions, as AGP replacements in animal feed. Four of these *Bacillus* strains (BPR-11, BPR-14, BPR-16 and BPR-17) showed antimicrobial activity against the pathogenic strains *Clostridium perfringens, Escherichia coli* and *Staphylococcus aureus* at 25 μg/mL, with BPR-16 and BPR-17 also able to inhibit *Pseudomonas aeruginosa* and *Salmonella enterica* at 100 μg/mL. Further chemical investigation of BPR-17 led to the identification of eight metabolites, namely C16, C15, C14 and C13 surfactin C (1–4), maculosin (5), maculosine 2 (6), genistein (7) and daidzein (8). Purified compounds (1–4) were able to inhibit all the tested pathogens with MIC values ranging from 6.25 to 50 μg/mL. Maculosin (5) and maculosine 2 (6) inhibited *C. perfringens*, *E. coli* and *S. aureus* with an MIC of 25 μg/mL while genistein (7) and daidzein (8) showed no activity. An animal trial involving feeding BPR-11, BPR-16 and BPR-17 to a laboratory poultry model led to an increase in animal growth, and a decrease in feed conversion ratio and mortality. The presence of surfactin C analogues (3–4) in the gut following feeding with probiotics was confirmed using an LC–MS analysis. The investigation of these *Bacillus* probiotics, their metabolites, their impacts on animal performance indicators and their presence in the gastrointestinal system illustrates that these probiotics are effective alternatives to AGPs.

## 1. Introduction

Antimicrobial growth promoters (AGPs) are currently utilised in the agricultural industry to improve livestock production, feed-energy conversion and to prevent the spread of infectious diseases [1,2,3]. These AGPs eliminate the microbiota in the gut and allow the host to access more nutrients [4,5]. They also halt the production of toxins produced by pathogens and improve livestock production [4,5]. However, there has been a steady rise of antimicrobial resistance due to the misuse of these AGPs. This has driven restrictions and bans of AGPs in countries throughout the EU, U.S. and Indonesia [6]. These regulations have been linked to decreases in livestock production and higher rates of food-borne infections, highlighting a need for safer and more effective alternatives to AGPs, such as probiotics [6,7,8,9].

Probiotics are live microorganisms that provide a range of benefits to their hosts once consumed [10,11]. Their benefits include the production of enzymes that assist in breaking down indigestible material, providing competition for nutrients in the gut to inhibit pathogenic bacterial growth and the production of antimicrobial metabolites [12,13,14,15,16]. These mechanisms are linked to an observed increase in animal growth and a reduction in gastrointestinal diseases and infection [17]. However, the research and strategies regarding probiotic use are often optimised to relatively few bacteria, as the majority of marketed probiotics stem from the lactic acid bacteria group [18]. Whilst the use of lactic acid bacteria has been generally accepted as being safe for consumption, these bacteria are typically anaerobic [19]. This makes commercial production difficult, as it requires utilising equipment and procedures to ensure that the bacteria are not exposed to the aerobic environment, leading to the release of toxic by-products [20]. Additionally, lactic acid bacteria are vulnerable to external environmental conditions such as temperature and acidity, making them unsuitable for pelleting processes and unsuitable for long-term storage. Thus, researchers have recently focused on utilising spore-forming bacteria from the genus *Bacillus* [21].

*Bacillus* are Gram-positive, facultative aerobic probiotics that have been widely studied for their ability to form self-protecting shells [22]. These shells are made of a proteinaceous coat, consisting of S-layer proteins, aminopeptidases and flagellin [23]. This coating allows the bacteria to survive harsh external conditions to germinate in the gut, making them suitable for use in animal feed as probiotics [24]. Once sporulated in the gastrointestinal tract, these probiotics can release a wide range of active substances including enzymes, nutrients and antimicrobial compounds [25]. These metabolites promote a microbiome ecosystem that supports the growth of these probiotics, whilst preventing the growth of toxin-producing microbes [26]. Nevertheless, the isolation and characterisation of the bioactive metabolites are rarely performed. This obstructs the progress in further improving bioactivity that could be carried out by discovering interactions or synergistic effects between different compounds [27]. Furthermore, it is uncertain whether the antimicrobial compounds produced by laboratory fermentation are also produced in the gastrointestinal environment during normal metabolism [27].

In this study, six *Bacillus* strains, namely BPR-11, BPR-12, BPR-13, BPR-14, BPR-16 and BPR-17, were evaluated against the pathogens *Clostridium perfringens*, *Escherichia coli*, *Pseudomonas aeruginosa*, *Staphylococcus aureus* and *Salmonella enterica*. These bacteria were selected for study as they all negatively impact broiler chicken production globally [28]. The antimicrobial activity warranted further chemical investigation to identify bioactive secondary metabolites. The animal trial on broiler chickens supplemented with the *Bacillus* strains were conducted, and animal performance and the presence of antimicrobial secondary metabolites in the gut were assessed.

## 2. Results

### 2.1. The Antimicrobial Activity of Selected Bacillus Strains

Six *Bacillus* strains (BPR-11, BPR-12, BPR-13, BPR-14, BPR-16 and BPR-17) were selected (Table 1) for the current study. Each strain was cultured in a tryptone soy broth (TSB) medium. The bacterial cultures were then lysed, centrifuged and the cell pellet removed. The supernatant broth was split into two halves: the first half was extracted with ethyl acetate (EtOAc) to give the EtOAc extract while the second half was freeze dried and resuspended in water to provide an aqueous crude extract.

The antimicrobial activity of the EtOAc extracts was evaluated against *C. perfringens*, *E. coli*, *P. aeruginosa*, *S. aureus* and *S. enterica* at the concentrations of 200, 100, 50 and 25 µg/mL (Appendix A). The results showed that the EtOAc extracts of strains BPR-11, BPR-14, BPR-16 and BPR-17 inhibited *C. perfringens*, *E. coli* and *S. aureus* growth at 25 μg/mL, whilst the EtOAc extracts of BPR-16 and BPR-17 were also active against *P. aeruginosa* and *S. enterica* at 100 μg/mL. The EtOAc extracts of BPR-12 and BPR-14 showed no activity. The crude extracts of the *Bacillus* strains were tested for antimicrobial activity at 800, 400, 200 and 100 μg/mL. However, no activity was detected for any of the crude extracts at the concentrations tested.

### 2.2. Isolation and Structure Elucidation of Antimicrobial Compounds

The chemical compositions of the active EtOAc extracts from BPR-11, BPR-14, BPR-16 and BPR-17 were analysed by HPLC (Figure 1B–E) and ^1^H NMR (Appendix A). They were also compared with the EtOAc extract of the culture media TSB (Figure 1A, Appendix A). The chromatograms showed similar profiles among BPR-11, BPR-14, BPR-16 and BPR-17, comprising an initial peak inherent from TSB at 0–2 min and a major peak at 4.5 min indicative of secondary metabolites. BPR-17 (Figure 1E) also contained several minor compounds at the retention time between 4–6 min. As BPR-17 was the most active and most abundant in secondary metabolites, it was selected for further chemical investigation to identify the antimicrobial secondary metabolites.

The broth of large-scale fermentations was extracted with EtOAc and the dried extract was purified by repeated HPLC. First, the EtOAc extract was fractionated on a C_18_ column with H_2_O/MeOH/0.1% TFA gradient elution, resulting in the collection of ten fractions, based on the elution of compounds (Figure 2). The activity of each fraction was assessed at a single dose of 100 µg/mL. Fraction 7 and 9 exhibited activities against *C. perfringens, E. coli* and *S. aureus*, and fraction 9 was also active against *P. aeruginosa* and *S. enterica* (Figure 2). These two fractions were, therefore, selected for further purification.

Fraction 9 was first purified by a normal phase Diol column (150 × 20 mm) with gradient elution using hexane/isopropanol, resulting in 4 surfactin C type lipopeptides, namely C16 (1, 0.3%), C15 (2, 1.3%), C14 (3, 0.8%) and C13 (4, 0.5%) surfactin C [29] (Figure 3). Fraction 7 was purified by a Luna C_18_ column (250 mm × 10 mm) with H_2_O/MeOH/0.1% TFA gradient elution, yielding maculosin (5, 1.0%) [30], maculosine 2 (6, 0.6%) [31], genistein (7, 0.5%) [32] and daidzein (8, 0.3%) [33] (Figure 3). The chemical structures of 1–8 were determined by ^1^H, COSY, HSQC, HMBC NMR and MS spectroscopic data (Appendix A) as well as optical rotations. They were identical to those reported in the literature [34,35].

The antimicrobial activity of compounds 1–8 and the minimum inhibition concentration (MIC) was measured at the concentrations of 1.563–200 µg/mL with a series of 1/2 dilutions. The results (Table 2) suggested that surfactin C analogues (1–4) were the most active, showing activity against *E. coli* (MIC = 6.25 µg/mL), *S. aureus* (MIC 6.25–12.5 µg/mL)*, C. perfringens* (MIC = 12.5 µg/mL), *P. aeruginosa* (MIC = 25 µg/mL) and *S. enterica* (MIC = 25 µg/mL) (Table 2). Maculosin (5) and maculosine 2 (6) were active against *C. perfringens, E. coli* and *S. aureus* (MIC values of 25 µg/mL), but inactive against other pathogens at the concentration of up to 200 µg/mL. Flavonoids (7, 8) showed no activity at the tested concentrations.

### 2.3. Animal Trial

The animal trial was conducted on 256 broiler hatchlings of mixed sex for 28 days. The hatchlings were individually weighed and birds that had a similar weight (±2 g) were randomly assigned to either a control or probiotic group. The control group was fed a basal, medication free wheat–soybean diet, whilst the probiotic group’s feed was supplemented with a probiotic mixture F1, consisting of BPR-11, BPR-16 and BPR-17 (Table 3). The broiler chickens were reared in floor pens and had ad libitum access to feed and water for 28 days [36].

The body weight (BW), average daily gain (ADG), average daily feed intake (ADFI) and feed conversion ratio (FCR) were recorded on a weekly basis. The data collection after 28 days is shown in Table 4.

The *Bacillus* probiotic-treated groups showed an overall increase in the measured indicators. Specifically, an average increase in mean BW (1573 g to 1606 g; *p* = 0.12), ADG (56.19 g to 57.36 g; *p* = 0.12), ADFI (71.95 g to 72.47 g; *p* = 0.63) and a decrease in FCR (from 1.28 to 1.26; *p* = 0.01) (Table 4) were observed for 28 days. Although the treated group was not significantly different from the control for the first 14 days, a significant increase was evident between day 14–21 in BW (885.46 g to 910.57 g; *p* = 0.04), ADG (66.99 to 70.05; *p* = 0.005) and ADFI (83.49 g to 87.58 g; *p* = 0.00061). A significant decrease in FCR (0–28 days) was detected when fed with probiotics (*p* = 0.01), indicating a significantly improved conversion of feed to energy levels. Lastly, the probiotic treatment group showed a lower mortality rate (1.56%) compared with the control group (3.13%).

### 2.4. Caecum Content Analysis

To evaluate whether the antimicrobial metabolites were present in the animal gut, caecum samples were collected from control and probiotic-supplemented birds for targeted metabolomic analysis. At the end of the animal trial, one bird per pen was euthanised by CO_2_ chamber and eviscerated to collect the caecum. Only male birds with similar body weights were selected. The samples collected from the trial were stored at –80 °C before being processed. Each sample was extracted with EtOAc/H_2_O (1:1), then vortexed and centrifuged. The EtOAc layer was dried and redissolved in MeOH and the MeOH solution was analysed by LC–MS. Ions, representing compounds 1–8, were extracted on both positive and negative ionisation modes using the molecular weights of the compounds. Fifteen gut samples were randomly selected from control and probiotic treated group.

For the probiotic treated group, surfactin C analogues (3–4) were present in 12 out of 15 samples (Figure 4A,B). Compounds 1, 2, 5 and 6 were also detectable (Appendix A); however, the signals were weak compared to the noise. For the control group, none of the compounds were detected in any of the 15 samples.

## 3. Discussion

*Bacillus* probiotics are renowned for their ability to produce a range of antimicrobial compounds, hence they act as alternative AGPs [37]. In this study, we selected six *Bacillus* strains, investigated their antimicrobial potential, secondary metabolites and their effects on animal growth [38]. Our results showed that four strains, BPR-11, BPR-14, BPR-16 and BPR-17, inhibited the activity of *C. perfringens, E. coli* and *S. aureus*, with BPR-16 and BPR-17 also inhibiting *P. aeruginosa* and *S. enterica*. Further chemical analysis on BPR-17, one of the most active strains, led to the isolation of antimicrobial lipopeptides (1–4) and dipeptides (5–6). Subsequent animal trials involving the feeding of a probiotic mixture including BPR-11, BPR-16 and BPR-17 over 0–28 days showed an increase in BW, ADG and ADFI and a decrease in FCR and mortality. By analysing the caecum samples from the animal trials, we were able to detect signals indicative of lipopeptides (3–4). The results suggest that antimicrobial surfactins may be responsible for the beneficial effects shown in the animal trials, supporting *Bacillus* probiotics as an effective alternative to AGPs.

The selection of strains for the animal trials was based on multiple factors, including the antimicrobial potential and the compatibility in fermentation. BPR-17 and BPR-16 were the most active followed by BPR-11 and BPR-14. Co-incubation of the mixed strains with pathogens suggested that the combination of BPR-17, BPR-16 and BPR-11 gave the best activity against all tested pathogens (unpublished results). In addition, the benefit of having multiple strains is well known in the feed industry, as there is an increasing tendency to work with multi-strain probiotics [39].

The most potent antimicrobial metabolites revealed in this study were the lipopeptides C16–C13 surfactin C (1–4). These metabolites were first reported in 1968 as products of *Bacillus subtilis* and are renowned for their potency to target both Gram-positive and negative strains [29]. These lipopeptides are characterised by a peptide moiety consisting of the seven amino acids (L-Glu, L-Leu, D-Leu, L-Val, L-Asp, D-Leu and L-Leu) with a β-hydroxy fatty acid tail and act in a concentration-dependent manner [40]. Previous studies showed that these lipopeptides form unilamellar vesicles on the membrane wall, leading to cell distortion and triggering apoptosis [41,42,43,44]. These compounds can also form pores at the plasma membrane that lead to intracellular leakage and eventual cell death [42,45,46,47]. Additionally, surfactins can breakdown bacterial biofilms, either by reducing the ratio of alkali-soluble polysaccharides or down-regulating the expression of genes involved in biofilm formation [48]. In our study, we found that the length of the lipophilic fatty acid affects its antimicrobial activity. This finding is consistent with the literature, showing that an extended fatty acid improves its ability to permeabilise the bacterial membrane leading to cell death [49].

Additionally, maculosin (5) and its analogue maculosine 2 (6) were isolated from the culture of BPR-17. Maculosin, a dipeptide, has been previously reported to exert an antimicrobial effect against methicillin-resistant *S. aureus* (MRSA) (37.0 µg/mL), *E. coli* (37.0 µg/mL) and *C. albicans* (27.0 µg/mL) [30,50]. The mechanisms of maculosine 2 have been purported to be due to its interaction with lipid heavy cell walls eventually leading to cell death. Furthermore, maculosin has been shown to antagonise quorum sensing, which is needed for bacterial interaction and growth [51,52]. In comparison, the activity of compound 6 has been reported to be lower compared to 5, highlighting the importance of the phenol group for its antimicrobial activity [30,53].

Genistein (7) and daidzein (8) have been identified for their antioxidant properties and have been explored for use in animal feed to improve animal health [32,54,55,56]. In our study, we were able to isolate these two isoflavones, but there was no antimicrobial activity detected for concentrations up to 100 µg/mL [57]. Although genistein has been previously reported to inhibit *S. aureus* and MRSA after only a limited incubation of 5–10 h, the authors noted that this inhibition was only a short term inhibition of topoisomerase IV and, after 10 h, the genistein was degraded by the bacteria in the culture, leading to a lack of inhibition over extended periods [58]. Additionally, daidzein has been shown to inhibit topoisomerase I, and, more weakly, topoisomerase II [59]. These actions halt the nucleic acid system, showing that these compounds are bacteriostatic rather than bacteriocidic as observed with our antimicrobial testing [60].

This study also investigated the effects of *Bacillus* probiotics on animal performance indicators and their faeces. Increases in BW, ADG and ADFI and a decrease in FCR were observed and have been associated with the altered fermentation and microbial ecology in the gut [61]. Similar benefits to animal performance has also been observed by Cavazzoni et al. (1998) [62]. Here, the addition of *B. coagulans* probiotics to the diets of poultry promoted growth and had prophylactic effects compared to chickens fed without additives. Additionally, the benefits to animal performance were comparable to animals fed with the antibiotic virginiamycin, commonly used as a growth promoting agent. These results can be explained by assuming that the *Bacillus* supplementation enhanced the feed utilization and reduced the excretion of proteins in the faeces [63]. Previous research has also highlighted the reduction of NH_3_-N of animals fed with *Bacillus* probiotics, suggesting a microbial aspect that reduces the production of urease [64]. This has been reinforced with recent studies, which have investigated the ability of *Bacillus* probiotics to affect the microbial community and to improve animal performance [65,66]. Here, not only could *Bacillus* probiotics competitively exclude pathogenic bacteria, but they were able to hinder the growth of the pathogenic bacteria *E. coli* and *S. enterica*, leading to an improvement in overall animal health [67,68,69].

Research into the use of probiotic interventions and their exact influence on the microbial communities using nucleic acid sequencing has increased over the past decade [70]. However, these changes in the microbial communities are often indirect and have resulted in closer analyses of the metabolome itself. Targeted metabolomic analyses of the faecal contents in the caecum confirmed the presence of secondary metabolites (3–4) in the gut. This follows previous findings, showing that probiotic supplementation changes the metabolite profiles in the gut using GC–MS and ^1^H NMR [71]. *Lactobacillus* probiotic supplementation was able to increase the amount of health enhancing compounds such as amino acids, carbohydrates and lipids [67,72]. This includes glutamine and 5-aminoimidazole-4-carboxamid, produced by *L. acidophilus,* which increases enterocyte proliferation, regulation of tight junctions and suppression of inflammatory signalling pathways [72,73]. However, the potential for these antibacterial components was unexplored in this study, and they may be synergistic in their antimicrobial effects [74]. Additionally, the gut microflora have been shown to interact heavily with the blood metabolites, and further metabolomic analysis of either the serum, saliva or urine will help deconvolute this network [75]. These directions will help identify potentiators, and reveal how these metabolites interact with each other, and hence assist in optimizing the performance and benefits of probiotic supplementation [76].

## 4. Materials and Methods

### 4.1. General Experimental Procedure

The tryptone soya broth (TSB), mueller hinton broth (MHB), Tryptic Soy Agar (TSA) and Mueller-Hinton Agar (MHA) were purchased from Oxoid Australia (Thebarton, Australia). The indicator strains *C. perfringens* (ATC13124), *E. coli* (ATCC 43887), *P. aeruginosa* (ATCC 10145), *S. aureus*, (ATCC 6538) and *S. enterica* (ATCC 6960) were purchased from Invitrogen (Melbourne, Australia). The anti-foam C (polydiethylsiloxane) and resazurin were purchased from Sigma Aldrich (Melbourne, Australia). The broiler eggs (ROSS 308) were purchased from a commercial hatchery (Woodlands Hatchery, Australia) and were transferred to the Central Queensland Innovation and Research Precinct (CQIRP) in Rockhampton, Australia. A basal medication-free wheat-soybean diet was purchased from a commercial feed miller, Laucke Mills (Daveyton, Australia) and transported to CQIRP. Solvents for extraction and purification were HPLC grade from RCl Labscan (Gillman, Australia). The water was deionised and filtered through a Millipore Milli-Q PF system (0.45 µm).

NMR spectra were obtained using a Bruker Ascent 800 MHz spectrometer coupled with a TCI CryoProbe at 298 K in deuterated dimethyl sulfoxide (DMSO-*d_6_*) (Cambridge Isotope Laboratories, Tewksbury, MA, USA). The optical rotation was recorded using a Jasco P-1020 polarimeter using a quartz cell with 10-cm path length. The LC–MS analysis was conducted on a Thermo Scientific MSQ Plus single quadrupole ESI mass spectrometer equipped with an Accucore C_18_ LC column (2.6 μm, 150 mm × 2.1 mm) with H_2_O/MeOH/formic acid as the solvent system. A Themo Scientific Dionex Ultimate 3000 RS UHPLC system equipped with a pump, autosampler, diode array detector and automated fraction collector was used for chemical analysis and HPLC fractionation. A Jasco V-650 Spectrophotometer was used for optical density (OD) determination. A Biotek Spectre 2 microplate reader was used for the antimicrobial activity readout. The eggs were incubated in a Brinsea Ova-Easy 380 Egg Incubator in the animal trial.

### 4.2. Bacillus Strains and Fermentation

Six *Bacillus* strains (BPR-11, BPR-12, BPR-13, BPR-14, BPR-16 and BPR-17) were obtained from Bioproton Pty Ltd. (Brisbane, Australia). The *Bacillus* strains were streaked on a plate with TSA (1.5% agar) and incubated for 16 h. Subsequently, a single colony was isolated and incubated for 24 h in TSB (1 L) with vigorous shaking (300 rpm). A starting inoculation culture was prepared with 0.1 OD at 600 nm. The sample was inoculated into a culture broth of TSB (1 L) added to a Eppendorf bioreactor (1.5 L). The culture was incubated for 8 h at 37 °C, 300 rpm, pH 7, with O_2_ levels at 300 ppm. Antifoam C was added to remove excess foam from the bioreactor. The fermented cultures were stored at −20 °C until ready for extraction.

### 4.3. Compound Isolation

#### 4.3.1. Extraction

The culture broth (1 L) was sonicated three times for 5 min and centrifuged at 4 °C at 8000 rpm. The cell pellet was separated from the broth and stored at −20 °C. Half of the broth (500 mL) was extracted with EtOAc (500 mL) three times and the combined EtOAc extracts were brought to dryness using a rotary evaporator to obtain an EtoAc extract. The second half of the broth (500 mL) was freeze-dried and resuspended in H_2_O to give the crude extract.

#### 4.3.2. Isolation

The chemical analysis of the EtOAc extracts (1 mg/mL) was performed on a Phenomenex Onyx Monolithic C_18_ column (100 mm × 4.6 mm) with a gradient elution from 10% H_2_O/90% MeOH/0.1% TFA to 100% MeOH/0.1% TFA within 7 min, then back to 10% MeOH/90% H2O/0.1% TFA within 4 min, at a flow rate of 4 mL/min. For large-scale isolation, the EtOAc extract (500 mg) was loaded onto an Alltech Hyperprep PEP C_18_ column (250 mm × 22 mm) eluting with a solvent gradient from 10% MeOH/90% H_2_O/0.1% TFA to 100% MeOH/0.1% TFA within 50 min then kept at 100% MeOH/0.1% TFA for 10 min, with a flow rate of 9.0 mL/min. Sixty fractions were collected based on time (1 fraction per min) and the tubes were combined into 10 fractions based on their UV peaks for further antimicrobial testing.

#### 4.3.3. Purification

Fraction 9 was purified using a YMC diol-120 normal phase column (150 mm × 20 mm) with gradient elution from 100% hexane to 80% hexane/20% isopropanol within 60 min at a flow rate of 9 mL/min, leading to the isolation of C16 surfactin C (1, 0.3%), C15 surfactin C (2, 1.3%), C14 surfactin C (3, 0.8%) and C13 surfactin C (4, 0.5%). Active fraction 7 was purified on a Phenomenex Luna C_18_ column (250 mm × 21.2 mm) with a gradient elution from 10% MeOH/90% H_2_O/0.1% TFA to 100% MeOH/0.1% TFA within 60 min at a flow rate of 3.5 mL/min, yielding maculosin (5, 0.7%), maculosine 2 (6, 0.6%), genistein (7, 0.4%) and daidzein (8, 0.4%).

The following details are available in the Appendix A:

 C16 surfactin C (1): white powder; [α]_D_^23.0^ +7.3 (*c* 0.1, MeOH); ^1^H NMR and ^13^C NMR data see Appendix A; LRESIMS, ^1^H, COSY, HSQC and HMBC NMR spectra see Appendix A; C15 surfactin C (2): white powder; [α]_D_^23.0^ +7.0 (*c* 0.1, MeOH); ^1^H NMR and ^13^C NMR data see Appendix A; LRESIMS, ^1^H, COSY, HSQC and HMBC NMR spectra see Appendix A; C14 surfactin C (3): white powder; [α]_D_^23.0^ +7.8 (*c* 0.1, MeOH); ^1^H NMR and ^13^C NMR data see Appendix A; LRESIMS, ^1^H, COSY, HSQC and HMBC NMR spectra see Appendix A; C13 surfactin C (4): white solid; [α]_D_^23.0^ +8.1 (*c* 0.1, MeOH); ^1^H NMR and ^13^C NMR data see Appendix A; LRESIMS, ^1^H, COSY, HSQC and HMBC NMR spectra see Appendix A; Maculosin (5): white solid; [α]_D_^23.0^ −20.1 (*c* 0.1, MeOH); ^1^H NMR and ^13^C NMR data see Appendix A; LRESIMS, ^1^H, COSY, HSQC and HMBC NMR spectra see Appendix A; Maculosine 2 (6): white solid; [α]_D_^23.0^ −16.0 (*c* 0.1, MeOH); ^1^H NMR and ^13^C NMR data see Appendix A; LRESIMS, ^1^H, COSY, HSQC and HMBC NMR spectra see Appendix A; Genistein (7): brown powder; ^1^H NMR and ^13^C NMR data see Appendix A; LRESIMS, ^1^H, COSY, HSQC and HMBC NMR spectra see Appendix A; Daidzein (8): brown powder; ^1^H NMR and ^13^C NMR data see Appendix A; LRESIMS, ^1^H, COSY, HSQC and HMBC NMR spectra see Appendix A.

### 4.4. Antimicrobial Assay

Bacteria were streaked on plates containing MHA (28 g/L, 1% Agar). A colony was picked, added to MHB and incubated at 37 °C until a starting culture of 0.1 OD was obtained. To test for activity, bacterial broth (78 µL) was added to either the crude extracts (800, 400, 200 and 100 µg/mL, 2 µL) or the EtOAc extracts (100, 50, 25, 12.5 µg/mL, 2 µL). The plate was then incubated at 37 °C overnight. A resazurin dye (25 µL) was then added into each well and incubated at 37 °C for 6 h, and the antimicrobial activity was quantified by the optical density at 525/580 nm on a plate reader. The serial dilutions were prepared with DMSO. Gentamicin (Sigma-Aldrich) was used as a positive control and DMSO was used as a negative control. Each concentration was tested in triplicate (*n* = 3).

For MIC determination of the pure compound, a range of concentrations were prepared (200, 100, 50, 25, 12.5, 6.25, 3.13 or 1.563 µg/mL, 2µL) and the assay was conducted as described for the extracts. The MIC was determined by the inhibition of visual growth of bacterium.

### 4.5. The Animal Trial

This study used a subset of an animal trial approved by the Ethics Committee of Central Queensland University under approval number 0000023123. A total of 256 mixed-sex broiler eggs were randomly selected and placed into two identical incubators with automatic digital humidity control and temperature sensitivity down to 0.1 °C and were digitally monitored 24 h per day for 21 days of incubation. The incubators were set at 37.5 °C and 55% relative humidity (RH) on day 1–18, and 37.5 °C and 65% RH on day 18–21. The hatchlings were individually weighed and birds with similar weight (±2 g) were randomly assigned to one of two experimental groups (control and probiotic). Each experimental group had 16 replicate pens with eight birds in each pen (*n* = 128 per experimental group). The feed-supplemented probiotic was added to the basal diet in the facility, and the diet was thoroughly mixed prior to the start of the experiment. The experimental treatments included a standard micro crumble wheat–soybean diet (control) and control +500 g/t probiotics (F1). The F1 probiotic added to the feed was supplied by Bioproton Pty Ltd. (Acacia Ridge, Queensland, Australia) (Table 3).

The broiler chickens in each group were reared in floor pens (120 cm × 120 cm × 80 cm) with wood shavings as bedding and had ad libitum access to feed and water for 28 days. Throughout the trial period, the ROSS 308 Management Handbook was followed to meet the nutritional and environmental recommendations of the broiler, including temperature, relative humidity (RH) and lighting program [36]. The temperature was set at 32 °C and 40–50% RH for the first 7 days and a 2 °C reduction per week until the temperature reached 22 °C at 28 days.

The BWs of individual birds and the FIs of each replicate pen were recorded weekly. The ADG, ADFI and FCR ratio between feed intake over BW of birds per replicate were calculated. Animal mortality was recorded and the data was used to adjust subsequent measurements.

### 4.6. Targeted Metabolomic Analysis

The caecum samples from the animal trial were stored at −80 °C and were analysed following a protocol previously described by Martias [77]. The caecum samples were freeze-dried, and a portion of the samples (6 mg) was added to H_2_O (6 mL) and ethyl acetate (6 mL). Each suspension was vortexed for 5 min and centrifuged at 12,000× *g* for 10 min at 4 °C. Subsequently, the supernatant (400 µL) was removed, concentrated and redissolved in 100% methanol (30 µL). The samples were analysed by the LC–MS system with a gradient elution from 5% H_2_O/95% MeOH/0.1% formic acid to 100% MeOH/0.1% formic acid within 13 min, at a flow rate of 0.3 mL/min. Ions were extracted using both positive and negative modes.

## Figures and Tables

**Figure 1 antibiotics-12-00407-f001:**
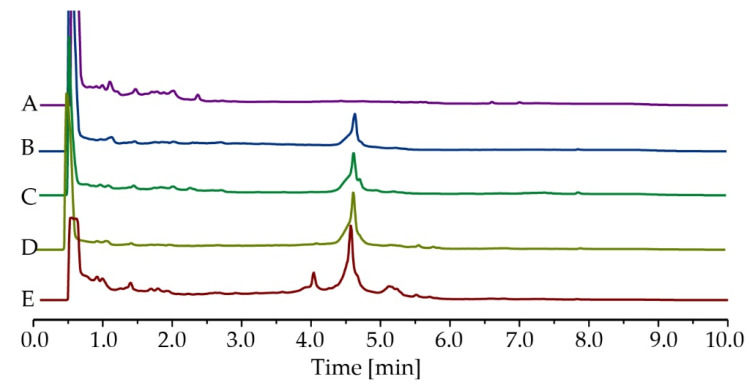
HPLC chromatograms of EtOAc extracts at 214 nm, (**A**) TSB, (**B**) BPR-11, (**C**) BPR-14, (**D**) BPR-16 and (**E**) BPR-17. Peaks at 0–2 min were from the culture medium TSB. Peaks at 4–6 min were metabolites produced by *Bacillus*.

**Figure 2 antibiotics-12-00407-f002:**
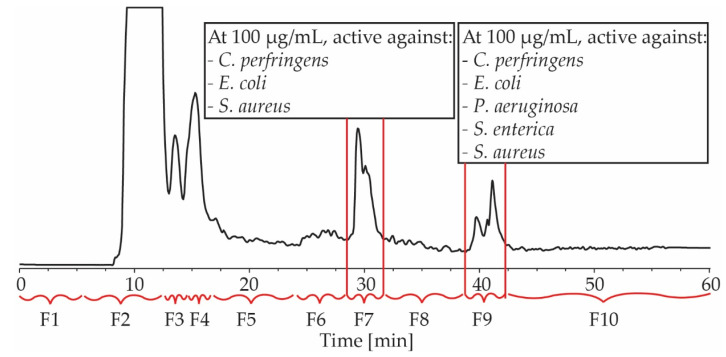
An HPLC chromatogram of BPR-17 EtOAc extract at 214 nm. Indicated are fractions 1–10 (F1–F10) and their antimicrobial activity at 100 µg/mL. F7 and F9 were active against *C. perfringens, E. coli* and *S. Aureus,* while F9 was also active against *P. aeruginosa* and *S. enterica*.

**Figure 3 antibiotics-12-00407-f003:**
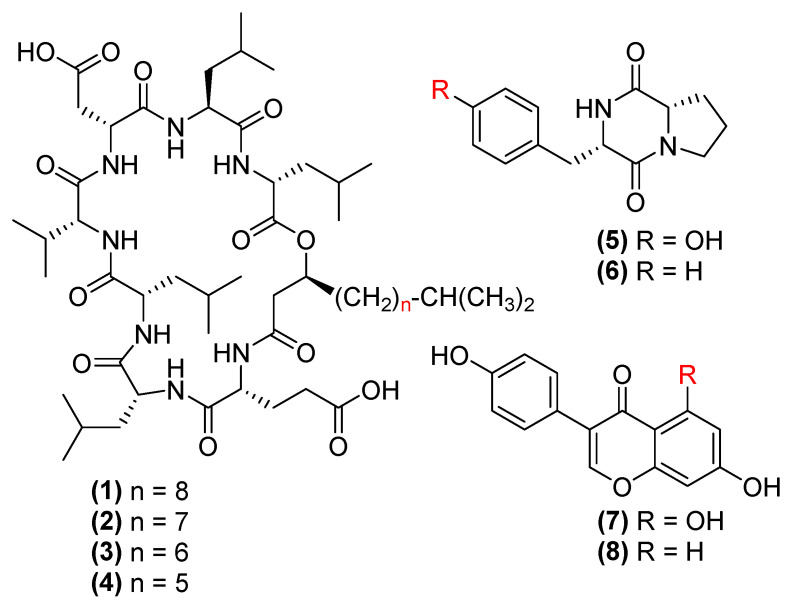
The chemical structures of C16-C13 surfactin C (1–4), maculosin (5), maculosine 2 (6), genistein (7) and daidzein (8).

**Figure 4 antibiotics-12-00407-f004:**
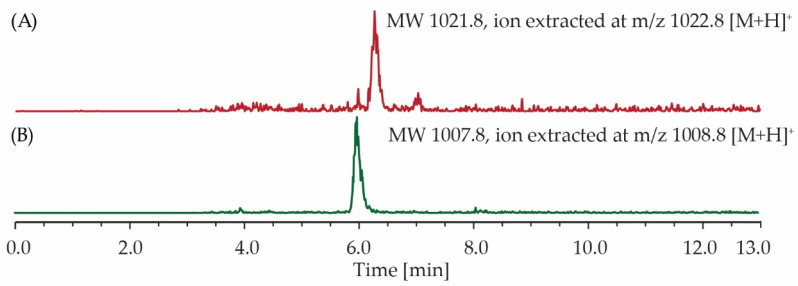
LC–MS ion extraction chromatograms of caecum extracts produced from a single animal sample fed with *Bacillus* composition F1. The spectra revealed the presence of (**A**) C14 surfactin C and (**B**) C13 surfactin C. MW—Molecular Weight.

**Table 1 antibiotics-12-00407-t001:** *Bacillus* strains, origins and CBS numbers.

Strains	Strain Type	Origin	CBS Number *
BPR-11	*B. amyloliquefaciens*	Soil and vegetation	141692
BPR-12	*B. licheniformis*	Soil and vegetation	141691
BPR-13	*B. subtilis*	Soil and vegetation	141693
BPR-14	*B. amyloliquefaciens*	Soil and vegetation	142360
BPR-16	*B. velezensis*	Soil and vegetation	148295
BPR-17	*B. amyloliquefaciens*	Soil and vegetation	148296

* *Centraalbureau voor Schimmelcultures* (CBS) number, generated by Westerdijk Fungal Biodiversity Institute, Utrecht, The Netherlands.

**Table 2 antibiotics-12-00407-t002:** MIC values of compounds 1–8 (µg/mL).

Samples	*C. perfringens*	*E. coli*	*P. aeruginosa*	*S. enterica*	*S. aureus*
C16 surfactin C	12.5	6.25	25	25	6.25
C15 surfactin C	12.5	6.25	25	25	6.25
C14 surfactin C	12.5	6.25	25	25	6.25
C13 surfactin C	12.5	6.25	25	25	12.5
Maculosin	25	25	NA	NA	25
Maculosine 2	25	25	NA	NA	25
Genistein	NA	NA	NA	NA	NA
Daidzein	NA	NA	NA	NA	NA

NA: not active at the tested concentrations.

**Table 3 antibiotics-12-00407-t003:** *Bacillus* strains and specifications in F1 probiotics.

Probiotic Strains	Specification
*Bacillus amyloliquefaciens* (BPR-11)	2 × 10^8^ CFU */g
*Bacillus amyloliquefaciens* (BPR-16)	2 × 10^8^ CFU */g
*Bacillus amyloliquefaciens* (BPR-17)	2 × 10^8^ CFU */g

* CFU—Colony Forming Units.

**Table 4 antibiotics-12-00407-t004:** The effect of the *Bacillus* mixture (F1) on performance parameters of broiler chickens from day 1 to 28 (*n* = 256). The orange highlights indicate statistical significance with *p* value ≤ 0.05.

	Treatment Group (*n* = 128)		*p* Value
	Control	Probiotic	
BW0 (g)	42.38 ± 0.26	42.97 ± 0.27	0.11
BW7 (g)	152.30 ± 1.57	155.91 ± 1.74	0.14
BW14 (g)	416.54 ± 4.38	420.24 ± 4.59	0.53
BW21 (g)	885.46 ± 9.47	910.57 ± 9.02	0.04 *
BW28 (g)	1573.27 ± 13.43	1606.12 ± 15.64	0.12
ADG0–7 (g)	15.70 ± 0.23	16.14 ± 0.24	0.23
ADG7–14 (g)	37.75 ± 0.58	37.76 ± 0.73	0.99
ADG14–21 (g)	66.99 ± 0.75	70.05 ± 0.66	0.005 *
ADG21–28 (g)	98.26 ± 1.55	99.36 ± 1.15	0.57
ADG0–28 (g)	56.19 ± 1.94	57.36 ± 2.14	0.12
ADFI0–7 (g)	16.47 ± 0.29	16.52 ± 0.21	0.90
ADFI7–14 (g)	46.50 ± 0.59	46.78 ± 0.61	0.74
ADFI14–21 (g)	83.49 ± 0.70	87.58 ± 0.80	0.00061 *
ADFI21–28 (g)	138.12 ± 1.67	138.69 ± 1.57	0.80
ADFI0–28 (g)	71.95 ± 2.00	72.47 ± 2.68	0.63
FCR0–7 (g/g)	0.76 ± 0.01	0.74 ± 0.00	0.17
FCR7–14 (g/g)	1.23 ± 0.01	1.24 ± 0.01	0.58
FCR14–21 (g/g)	1.25 ± 0.01	1.25 ± 0.00	0.70
FCR21–28 (g/g)	1.41 ± 0.01	1.40 ± 0.01	0.29
FCR0–28 (g/g)	1.28 ± 0.00	1.26 ± 0.00	0.01 *
Mortality (%)	3.13	1.56	

* Statistically significant using student unpaired *t*-test with *p* value ≤ 0.05.

## Data Availability

The data is contained in the manuscript and Appendix A.

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
