# Peer review of "Antimicrobial Properties of Bacillus Probiotics as Animal Growth Promoters"

_antibiotics, 2023, doi:10.3390/antibiotics12020407_

Round 1

Reviewer 1 Report

1. Title Should be: Antimicrobial Properties of Bacillus Probiotics as Animal Growth Promoters

2. Bacillus is only a genus name, not a species, and it doesn't have to be italicized.

3. Abstract is too long, authors must refer to the journal guideline. Please shorten, if possible a maximum of 300 words. That's current format you submitted are 330 words.

4. Provide a legend for each image and table used, for example in table 1 what CBS stands for must be given a description, etc. Table 4, P values marked with colors need further explanation. What does MW mean in figure 4? Information / legend must be given below the picture.

5. EtOAc and all initial abbreviations must be given the abbreviation where it appears first.

6. Writing n or indicating the meaning as the number of samples, it must be italicized.

Author Response

Please see attached responses

Reviewer 2 Report

The work has significant value in mankind. However, revision is needed to improve the manuscript.

Keywords

Arrange alphabetically.

Introduction

Line 34: APG--- AGP

Results

Many parts will be in the Materials and Methods section, like lines 84-85, 92-94, 100-103, 114-115, 150-151, etc. Therefore, I suggest rewriting the Results section.

Line 258: S. Enterica--- S. enterica

References

Check the name of Journals, e.g., line 434, 448, 455, 458, 461, etc.

Reviewer 3 Report

Dear authors

ml is better to be mL.

The writing of the manuscript is well.

The mechanisms of the compounds can be described in the discussion. 

best regards 

Author Response

Response to reviewer 3

Comment 1: ml is better to be mL.

Responses: Authors have revised ml to mL, l to L and ml to mL throughout the manuscript.

Comment 2: The mechanisms of the compounds can be described in the discussion.

Responses: Authors have taken the reviewer’s comment on board and revised the 1st paragraph of discussion in the revised manuscript.  

Bacillus probiotics are renowned for their ability to produce a range of antimicrobial compounds, hence act as alternative AGP [37]. In this study, we selected 6 Bacillus strains, investigated their antimicrobial potential, secondary metabolites and their effects on ani-mal growth [38, 39]. Our results showed that four strains, BPR-11, BPR-14, BPR-16 and BPR-17, inhibited the activity of C. perfringens, E. coli, S. aureus with BPR-16 and BPR-17 al-so inhibiting P. aeruginosa and S. enterica. Further chemical analysis on BPR-17, one of the most active strains, led to the isolation of antimicrobial lipopeptides (1-4) and dipeptides (5-6). Subsequent animal trials involving the feeding of a probiotic mixture including BPR-11, BPR-16 and BPR-17 over 0-28 days showed an increase in BW, ADG, ADFI and a de-crease in FCR and mortality. By analysing the caecum samples of the animal trials, we were able to detect signals indicative of lipopeptides (3-4). The results suggest that antimicrobial surfactins may be responsible for the beneficial effects shown in the animal trails, supporting Bacillus probiotics as an effective alternative to AGP.”

Reviewer 4 Report

Comments for authors

The manuscript entitled “Antimicrobial Bacillus Probiotics as Animal Growth Promoters” described the isolation, extraction, and identification of antimicrobial metabolites secreted by probiotic Bacillus strains. The animal trial was also included to investigate the performance of these strains as an animal growth promoter. However, some serious flaws should be adopted to improve this MS as follows.

1.     The result of the antimicrobial activity of extracts was not clear. From the result, 25 ug/mL of EtOAc extracts of BPR-11, 14, 16, and 17 can inhibit C. perfringens, E. coli, and S. aureus, and EtOAC extracts of BPR-16 and 17 at 100 ug/mL can inhibit P. aeruginosa and S. enterica. What about another part of each extract? Are they have an activity?

There was no result for both parts of BRP-12 and 13 extracts mentioned.

Line 97-98: “None of the crude extracts showed antimicrobial activity at the concentrations of up to 800 μg/ml”. It was not clear. What does it mean?

The fermented broth of each isolate should be tested to compare the activity along with both extract parts. It could be possible that the solvent or freeze-dried process might destroy or reduce the activity of the actual active compounds presented in fermented broth.

Kindly revise this part and present these results as Table.

2.     Why only BRP-17 was chosen for the NMR study? In the animal trial, the mixture from three probiotic cultures (F1 probiotics) was applied for animal trial. What are the criteria for selecting? This is a main serious flaw. I think the NMR study should be performed on all of them. Please clarify.

 3.     Highlight in Table 4 can be changed to the symbols and the description should be shown as a footnote.

 4.     Section 3.3 should be revised and divided into subsections 3.3.1 for extraction, 3.3.2 for purification by HPLC and column chromatography, and 3.3.3 for characterization via MS/NMR. The current version is ambiguous and unstructured.  

5.     The method of MIC test for determining the concentration of compounds 1-8 on indicator strains should be included.

Author Response

Please see attached responses

Round 2

Reviewer 4 Report

I have no further comments.